# *Opuntia Ficus-Indica* Peel By-Product as a Natural Antioxidant Food Additive and Natural Anticoccidial Drug

**DOI:** 10.3390/foods12244403

**Published:** 2023-12-07

**Authors:** Meriem Amrane-Abider, Mirela Imre, Viorel Herman, Nedjima Debbou-Iouknane, Fairouz Saci, Hafid Boudries, Khodir Madani, Hafida Merzouk, Abdelhanine Ayad

**Affiliations:** 1Centre de Recherche en Technologies Agroalimentaires, Route de Targa Ouzemmour, Campus Universitaire, Bejaia 06000, Algeria; meriem.amrane@crtaa.univ-bejaia.dz (M.A.-A.); khodir.madani@univ-bejaia.dz (K.M.); 2Department of Parasitology and Parasitic Disease, Faculty of Veterinary Medicine, University of Life Sciences “King Mihai I” from Timisoara, 300645 Timisoara, Romania; 3Department of Infectious Diseases and Preventive Medicine, Faculty of Veterinary Medicine, University of Life Sciences “King Mihai I” from Timisoara, 300645 Timisoara, Romania; viorel.herman@fmvt.ro; 4Department of Environment Biological Sciences, Faculty of Nature and Life Sciences, University of Bejaia, Bejaia 06000, Algeria; ecokarima@gmail.com; 5Biotechnology Research Center (C.R.Bt.), Ali Mendjli Nouvelle Ville UV 03, BP E73, Constantine 25000, Algeria; sacifairouz@gmail.com; 6Department of Food Sciences, Faculty of Nature and Life Sciences, University of Bejaia, Bejaia 06000, Algeria; hafid.boudries@univ-bejaia.dz (H.B.); hafimer2009@gmail.com (H.M.); 7Laboratory of Biomathematics, Biochemistry, Biophysics and Scientometrics, Faculty of Nature and Life Sciences, University of Bejaia, Bejaia 06000, Algeria

**Keywords:** *Opuntia ficus-indica* peel, microwave, antioxidant activities, margarine oxidative stability, anticoccidial activity, *Eimeria*

## Abstract

The present study was carried out to valorize the *Opuntia ficus-indica* (OFI) by-products by extracting and identifying their biochemical compounds and evaluating their antioxidant potential by in vitro activities (DPPH radical and FRAP), as well as their capacity to stabilize margarine oxidation (rancimat test). In addition, their in vitro anticoccidial effect on the destruction of *Eimeria* oocysts isolated from naturally infected chickens was also targeted. Microwaves and response surface methodology tools were used to extract the maximum amount of phenolic compounds (42.05 ± 0.46 GAE mg/g DW of total phenolic compounds in 90 s at 400 watts). Moreover, the effect of extraction factors was also studied. Eight phenolic compounds, including isorhamnetin, dihydrokaempferol, and kaempferol-3-O-rutinoside, were identified. The findings confirmed that OFI peel extract has strong antioxidant activities (DPPH radical, ferric reducing power). The rancimat test shows that OFI peel extract improves margarine stability by 3.2 h. Moreover, it has a notable destruction rate of *Eimeria* oocysts (30.06 ± 0.51%, LC_50_: 60.53 ± 0.38 mg/mL). The present investigation offers promise for the reuse of food waste as natural margarine additives, protection of the environment, and substitution of anticoccidial synthetic treatments.

## 1. Introduction

The growth of industries and technology has led to an increase in the amount of waste and by-products released into the environment. In recent years, the concept of a green economy has appeared in the technological world and has been adopted by the International Trade Union Federation. This economic model is based on rules and principles of sustainable development, including respect for the environment by limiting waste and recovering by-products [1,2,3]. *Opuntia ficus-indica* peels (OFI) are one of the most important by-products generated by the prickly pear cactus industry; their percentage can exceed 40% up to 55% of the whole weight of the fruit [4,5]. OFI peels are rich in proteins, vitamins, minerals, and carbohydrates (dietary fiber and pectin) as well as bioactive substances known for their source of beneficial properties. Numerous studies have been published that found that OFI peels have multiple physiological activities such as antioxidant, anti-inflammatory, antimicrobial, and anti-acetylcholinesterase [4,6,7,8].

The valorization of OFI peel by-products is an absolute necessity to limit waste and preserve the environment, and they can be used in a more profitable way in human and animal feed. The valorization process frequently targets bioactive compounds. This may be extracted using a variety of techniques; however, their yield and antioxidant strength heavily depend on many variables, including extraction time, type of solvent, solvent concentration, etc. [9]. Indeed, several studies were investigated on the optimization of phenolic compound extraction parameters [10,11,12]. One of the greatest green process extraction techniques is microwave-assisted extraction. The recovery of polyphenols from natural matrices makes extensive use of it because of its interesting advantages, which include lower solvent usage (even solvent-free extraction is possible), time, and energy consumption [13,14]. These advantages can be attributed to the particular microwave-assisted extraction mechanism, which consists of the transformation of electromagnetic energy into heat by ionic conduction and dipolar rotation through dipole inversions and displacement of charged ions present in the bioactive compounds and the solvent.

One of the biggest challenges for researchers is assigning value to these bioactive compounds after extraction. Some are studying them for therapeutic purposes, while others are researching them as potential alternative food additives. Several molecules and natural extracts from mango, rambutan, grape, banana, prickly pear, potato, and tomato peels have shown their antioxidant potential in biscuits, vegetable oils, pasta, mayonnaise, margarine, rinds, and other different food formulations [15,16,17]. Margarine is one of the most susceptible foods to oxidation, which causes rancidity. That is a serious problem dreaded by manufacturers because it detracts from margarine’s nutritional and sensory qualities while forming free radicals and peroxides that are harmful to health. As a result, margarine is receiving a great deal of attention in the search for natural antioxidant alternatives [13,18].

On the other hand, there is growing interest in the investigation of bioactive ingredients in medications and animal feed. In recent years, great attention has been accorded to the use of OFI peel as animal feed [19,20]. Badr et al.’s [21] investigation is one of the studies that has confirmed the positive impact of OFI peel addition in chicken feed in terms of improved daily growth, feed conversion ratio, meat quality, and economic efficiency. Moreover, multiple studies found that using medicinal herbs in chicken feed rather than artificial medications boosts the immune system, enhances growth performance, and perhaps even has anticoccidial properties [22,23]. Two of the most common signs of coccidiosis in broiler chickens are diarrhea and weight loss. Despite developments in veterinary care, this illness continues to be the principal contributor to chicken mortality and financial losses. Many studies are concentrating on the use of natural extracts to treat *Eimeria*, which is its primary cause [24,25,26]. Moreover, the ability of OFI flowers to destroy *Emeria* oocysts has been demonstrated by Amrane-Abider et al.’s [27] research. Therefore, in order to valorize OFI peel by-products and evaluate their antioxidant and anticoccidial potentials, the following strategy was adopted: I. Maximize and optimize the OFI peel phenolic compound extraction using microwave-assisted extraction and response surface methods (Box–Behnken design). II. Determine the phenolic profile and the antioxidant capacity of OFI peel optimum. III Study their use as natural antioxidants in margarine. IV. Study the effect of that peel extract on the destruction of *Eimeria* oocysts isolated from naturally infected chickens (in vitro anticoccidial).

## 2. Materials and Methods

### 2.1. Plant Material

*Opuntia ficus-indica (OFI)* fruit samples were harvested in Talandjast, Bejaia department, northeast Algeria. After washing the fruit, the peels were dehydrated in a freeze-dryer (Christ, Alpha 1-4 LD plus, made in Germany), then ground, sieved, and stored in a cool place at 4 °C.

### 2.2. Chemical Reagents

All chemicals were obtained from Sigma Chemical (Sigma–Aldrich GmbH, Tauftkirchen, Germany), with the exception of sodium carbonate, which was obtained from Biochem, Chemopharma (GA, USA), and Folin–Ciocalteu (Biochem, Chemopharma, made in Montreal, QC, Canada). Diclosol^®^ is available from Avico and the Arab Industry Veterinary Co. in Amman, Jordan.

### 2.3. Microwave-Assisted Extraction

The microwave-assisted extraction technique was used to extract bioactive molecules from OFI peels according to the Amrane-Abider, Nerin, Cannelas, Zeroual, Hadjal, and Louaileche [13] method. Briefly, one gram of OFI peel powder and a volume (milliliter) of ethanol were mixed. The mixture was irradiated using a microwave (NN-S674MF, Maxi power, Shenzhen, China) under various test parameter settings (Table 1). The extraction suspension was immediately chilled in cold water. The *Opuntia ficus-indica* peel extract was filtered after being centrifuged at 4500 rpm for 10 min (NÜVE, NF-800-R Model, Ankara, Turkey).

### 2.4. Determination of Phenolic Compound Contents and Antioxidant Activities

#### 2.4.1. Determination of Total Phenolic Content (TPC)

The phenolic compound (TPC) content of OFI peels was determined in accordance with Velioglu et al. [28]. To summarize, 1.5 mL of diluted (1/10) Folin–Ciocalteu reagent was mixed with 500 microliters of OFI extract. After five minutes, one thousand five hundred microliters of 6% sodium carbonate were added to the initial mixture. Prior to measuring the absorbance at 750 nm in comparison to a blank, the mixture was incubated in the dark at room temperature for one hour (h). Gallic acid served as the calibration curve reference. The results are shown as milligrams of gallic acid equivalent (GAE) per gram of dry weight (DW).

#### 2.4.2. Determination of Antioxidant Activity (DPPH Radical Scavenging Assay)

Using the radical scavenger 1,1-diphenyl-2-picrylhydrazyl (DPPH), the antioxidant activity was evaluated according to Molyneux [29]. Two hundred microliters of the OFI peel optimum extract were mixed with 1 mL of methanolic DPPH radical solution (60 µM). The decolorizing process was detected at 515 nm after 30 min. DPPH radical scavenging activity was calculated using the following equation:(1)$$\mathrm{DPPH}\;\mathrm{radical}\;\mathrm{scavenging}\;(\%)=\frac{A_{0}-A_{1}}{A_{0}}\;\times \;100$$
where A_0_ is the absorbance of DPPH**^·^** solution with the extraction solvent and A_1_ is the absorbance is the absorbance of the OFI peel optimum extract. The findings are given as 50% inhibition concentration (IC_50_) and are expressed as mg/mL.

#### 2.4.3. Ferric Reducing Power (FRAP)

The reducing power of the peel optimum extract was evaluated as the capacity of reducing Fe^3+^ to Fe^2+^ according to the method by Amarowicz et al. (2004) [30]. Phosphate buffer (0.2 M, pH 6.6) and potassium ferricyanide (C_6_N_6_FeK_3_, 1%) were mixed with equivalent quantities (2.5 mL), and then 1 mL of OFI peel optimum extract was added. The mixture was incubated at 50 °C for 20 min. Two-point-five milliliters of trichloroacetic acid at 10% were added. For ten minutes, the mixture was centrifuged at 1750× *g*. Five hundred microliters of ferric chloride solution (0.1%), 2.5 mL of distilled water, and a 2.5 mL aliquot of the top layer were combined. The absorbance was measured at 700 nm, and the result was expressed as a concentration that reduces the oxidized iron by half (IC_50_).

### 2.5. Phenolic Compound Profile

*Opuntia ficus-indica* peels’ optimal phenolic components profile was obtained by using ultrahigh-performance liquid chromatography electrospray ionization quadrupole time-of-flight mass spectrometry (UPLC-ESI-Q-TOF-MS) as described in Amrane-Abider, Nerin, Cannelas, Zeroual, Hadjal and Louaileche’s [13] investigation.

### 2.6. Margarine Preparation and Oxidative Stability

Margarines were prepared according to the recipe described by Amrane-Abider, Nerin, Cannelas, Zeroual, Hadjal and Louaileche [13]. The major ingredients in the margarine formula were the fatty phase (82%) (palm oil, sunflower oil, and equivalent hydrogenated soybean, emulsifier (lecithin), and β-carotenes (12 mg/kg)), the watery phase (18%) (milk, aroma (diacetyl), salt (0.60%), lactic acid (0.5 mL/kg)), and the preservative (potassium sorbate). Three types of margarines were prepared to test the effectiveness of OFI peel optimum extracts in preventing margarine oxidation: a control margarine without any antioxidants, a reference margarine with vitamin E as an antioxidant (100 ppm), and one with OFI peel optimum extracts as natural antioxidants at two concentrations of 50 ppm and 100 ppm. The latter extract was included following solvent evaporation using rotavapor (Rotavapor^®^ R-300, Buchi) and lyophilization. To analyze and estimate the ability of the antioxidant molecules found in optimum OFI peel to stabilize and inhibit margarine oxidation, the rancimat test was carried out. The rancimat method is an accelerated oxidation tool that measures the conductivity of oxidation products (volatile compounds) while providing the induction time that characterizes the oxidation stability of margarine. The oxidative stability of margarines was determined using the rancimat test according to the ISO International Standard [31] method. In summary, three grams of the sample were heated to the melting point and then introduced into an air oxidation flask. Test conditions were set (the temperature was 98 °C and the gas flow rate was 10 L/h). Sixty-five milliliters of distilled water were filled into the measuring cells.

### 2.7. Determination of the Anticoccidial Activity of OFI Peel Optimum Extract

Isolation of *Eimeria* species from naturally infected chickens, as well as purification and identification, were based on methods described by Carvalho et al. [32]. The proportions of *Eimeria* species investigated (*Eimeria tenella* 75%, *Eimeria acervulina* 10%, *Eimeria mitis* 4%, *Eimeria preacox* 4%, *Eimeria maxima* 4%, and *Eimeria brunetti* 3%) as anticoccidial treatment are indicated in the work by Amrane-Abider, Imre, Herman, Debbou-Iouknane, Zemouri-Alioui, Khaled, Bouiche, Nerín, Acaroz, and Ayad [27] on the OFI flower. *Briefly*, the parasite sporulation was carried out in a potassium dichromate solution (2.5% K_2_Cr_2_O_7_) under high humidity over 70%, and a temperature of 29 ◦C. The buffered saline solution was used in *Eimeria* oocyst purification (PBS, containing sodium chloride, potassium chloride, disodium-phosphate double hydrated, and potassium dihydrogen phosphate at respective percentages of 8%, 0.2%, 1.13%, and 0.2%). To avoid denaturation of the antibiotic (penicillin V 100 IU) and antifungal (fluconazole 17 mg/mL) agents, sterilization was achieved by membrane filtration (diameter 0.2 µm at a pH of 7.4).

The HBSS (Hanks’ Balanced Salt Solution) medium is based on a mixture of salt (sodium chloride, potassium chloride, and calcium chloride in the following percentages: 8%, 0.4%, and 0.139%), with the addition of a carbohydrate source (1% dextrorotatory glucose) and complete with disodium phosphate, potassium dihydrogen phosphate, and magnesium sulfate (0.0478, 0.06, and 0.097%, respectively).

The in vitro anticoccidial activity of the OFI peel optimum extract was determined according to Debbou-Iouknane, Nerín, Amrane-Abider, and Ayad [22].

The effect of the OFI peel optimum extract on oocyst reduction was carried out as follows: two hundred microliters of peel extract were added to 1 mL of suspension (containing 100 µL of washed suspension of *Eimeria* oocysts at 24.5 × 10^5^ oocysts/mL) and 700 μL of phosphate-buffered saline. After incubating the parasite suspension and extracting it from the plant matrix in the dark at room temperature for the period 1.3.5.7 and 24 h, the samples were centrifuged at 320 g for 5 min and the absorbance of the supernatant was measured at 273 nm with a spectrophotometer.

The percentage of destroyed sporulated oocysts was then calculated using the following equation:(2)$$Nr=100-(\frac{Nt\times 100}{N0})$$
where: N_r_ = oocyst number reduction rate; N_t_ = oocyst number at x time (1 to 24 h); N_0_ = oocyst number at time 0 (24.5 × 10^5^ oocysts).

Diclazuril as a synthetic anticoccidial was evaluated at a concentration of 10 mg/mL under the same experimental conditions. The regression curve’s lethal concentration (LC_50_) for the diclazuril and OFI peel values was then deduced.

### 2.8. Experimental Design and Statistical Analyses

In order to maximize the extraction of phenolic compounds from OFI peels, a three-level, three-factor Box–Behnken (BBD) experimental design of the JMP program (version 10.0, SAS package) was used. Based on the literature and preliminary studies, three factors were optimized: X_1_ for ethanol concentration (40–80%), X_2_ for extraction time (30–90 s), and X_3_ for microwave power (400–1000 watts) (Table 1). All the experiments were performed in triplicate, and by using analysis of variance (ANOVA), a 95% confidence interval was set. The results were expressed as the mean value ± standard deviation (SD). Statistical analysis was performed using the STATISTICA 12 program.

## 3. Results and Discussion

### 3.1. Phenolic Compounds

In the present study, we investigated the effects of microwave variable extraction on the total phenolic content of OFI peels and identified their optimum combination. In addition, we evaluated the effect of the OFI peel optimum extracts on margarine oxidative stability as a natural antioxidant and their in vitro anticoccidial effect on the destruction of *Eimeria* oocysts (chicken parasites).

Investigations into food processing technologies should include effects on phenolic compounds and/or parameters that affect the stability of phenolic compounds in foods. Our results revealed that the experimental data of the Box–Behnken design matrix and the predicted values for the three independent variables affected the extraction of total phenolic components from the OFI peel, as shown in Table 1.

The current investigation shows that OFI peel extracts are highly rich in polyphenols. However, their content depends strongly on the extraction conditions; it varied from 12.59 to 40.11 mg GAE/g DW. Moreover, there is a good agreement between the experimental results and those predicted by the polynomial model. That was confirmed by the model’s coefficient of determination (R^2^) value, which is in the order of 0.97. While the adjusted value of the coefficient of determination was equal, R^2^_adjusted_ = 0.90, as shown in Table 2. The statistical analyses in Table 2 demonstrate that the model was significant (*p* < 0.01) and the lack of fit was not significant (*p* > 0.05), thereby confirming the validity of the model.

Table 3 illustrates the interaction between the ethanol concentration, extraction time, and microwave power factors on the extraction efficiency of OFI peel total phenolic compounds using microwave extraction. The concentration of ethanol as an independent variable significantly influences microwave extraction (*p* = 0.00089).

In addition, the ethanol concentration quadratic factor shows a significant (*p* < 0.00625) impact on bioactive extraction. On the other hand, the extraction time and power interaction has a strong impact on the extraction of bioactive chemicals from OFI peel.

The extraction time–-power interaction was by far the strongest factor interaction that had an impact on the extraction of the bioactive chemicals. Ethanol concentration–microwave power and extraction time–ethanol concentration interactions followed. The interaction between the factors in total phenolic compound extractions is well illustrated in Figure 1. According to Figure 1A, a low ethanol concentration in a short extraction time (4–30 s) does not allow a good extraction rate of phenolic compounds (7 mg GAE/g DW). Despite the prolongation of extraction time with low solvent concentrations, the extraction rate remains low (16 mg GAE/g DW), while it increases gradually with high concentrations. This result shows once again the importance of this factor in the extraction of bioactive compounds, which are known to be moderately polar since the ability of a solvent to extract compounds also depends on the physicochemical properties of the latter.

The same observation was found with the interactive influences of the ethanol concentration and microwave power extraction parameters (Figure 1B), where an increase in ethanol concentration increased the TPC extraction yield [33]. The high-power input facility of extraction of bio-available compounds becomes available quickly as a result of the solvent heating up quickly and the cell walls rupturing, increasing the amount of energy that is delivered to the substrate [34]. However, varying the microwave power alone (without varying the ethanol concentration) does not change the results much (15 mg GAE/g DW), whereas a simultaneous variation with the ethanol concentration changes the results significantly. The highest amount of phenolic compounds was obtained with 400 to 600 W and an ethanol concentration above 65 percent.

Moreover, time and microwave power extraction parameters were the most significant interactive factors, as shown in both Table 3 (*p* < 0.00077) and Figure 1C, which is in accordance with several studies [13,35]. Phenolic compound recovery is strongly dependent on microwave power, although it is often limited to a certain value [33]. On the other hand, exaggerated exposure to microwaves, even at low power levels, reduces extraction efficiency due to structural denaturation of bioactive compounds and oxidation [36]. Moreover, high microwave power at a minimum time may be at the root of biopolyphenol extraction efficiency [37]. It turned out that the inverse was also true; in both instances, the TPC level was higher than 40 GAE mg/g DW. Furthermore, the ideal condition for extracting a maximum TPC from OFI peels was low power (400 W) for 90 s using 67.86% ethanol, as shown in Figure 2. This result was confirmed experimentally by measuring TPC (42.05 ± 0.46 mg GAE/g DW). Our outcome was higher than several investigations conducted in Algeria, Egypt, and South Africa, where the TPCs of OFI peels were 28.68, 8.48, and 17.59 mg GAE/g DW, respectively [4,38,39].

The scavenging of free DPPH radicals by OFI peels was evaluated. This assay reflects the hydrogen donation ability of antioxidants. The result was expressed as an IC_50_ value, where a lower IC_50_ value indicated higher radical scavenging activity. *Opuntia ficus-indica* peel extract exhibits a high ability to scavenge DPPH radicals with an IC_50_ value of 12.99 ± 0.36 mg/mL (Table 4). This test not only confirms the antioxidant capacity of OFI peels but also confirms the mechanism of action of their compounds, which is to give hydrogen. Choe and Min [40] reported that natural antioxidants maintain their stability even after donating hydrogen, i.e., they become relatively stable antioxidant radicals with a low standard reduction potential, below 500 mV. Our result was higher than that found by Chougui et al. [41] (IC_50_ value of 77.81 mg/mL) for peel extract and Chougui et al. [42] (IC_50_ value of 49.60 mg/mL) for seed extract. On the other hand, 6.57 ± 0.05 mg/mL of the present extract reduces the oxidized iron by half. This result shows us another antioxidant mechanism of OFI peels, namely electron donation. The ferric-reducing power test is based on the ability of antioxidants to reduce ferric ions (Fe^3+^) to ferrous ions (Fe^2+^) [43]. On one side, that result was higher than that of Aruwa, Amoo, and Kudanga [39] (93.74 mg/mL for OFI peel); on the other side, it was lower than that of Chougui, Djerroud, Naraoui, Hadjal, Aliane, Zeroual, and Larbat [41] (1.03 mg/mL for OFI peel). These differences may be due to plant conservation, extraction processes, drying modes, and storage conditions, as reported in a recent literature review [44].

Phenolic profile analysis of OFI peel optimum by chromatography (UPLC-ESI-Q-TOF-MS) revealed the presence of eight phenolic molecules, including phenolic acids (vanillic acid, coumaric acid, and protocatechuic acid) and flavonoids (Table 5) (Figure 3).

Coumaric acid, dihydrokaempferol, and isorhamnetin were already identified by Amrane-Abider, Nerín, Tamendjari, and Serralheiro [4] and Chougui, Djerroud, Naraoui, Hadjal, Aliane, Zeroual, and Larbat [41] in OFI peel in Algeria. This last substance is a flavonol, frequently present in *Opuntia ficus-indica* as well as other plants, including *Hippophae rhamnoides* and *Ginkgo biloba* [45]. Isorhamnetin has a number of pharmacological properties, including cardiovascular and cerebrovascular protection, anti-inflammation, prevention of obesity, anti-tumor, anti-oxidation, anti-bacterial, and anti-viral [46]. The other compounds had already been identified in OFI fruit pulp and flowers [11,27,47,48,49,50]. The variability of phenolic compounds reported by various authors could be due to different extraction and analysis methods. In addition, maturity stages, geographical variations, and differences in culture or growth conditions could also play a role in the mentioned variability.

### 3.2. Margarine Oxidative Stability

The consumption of margarine has been rising gradually. However, due to its high fat content—around 80%—the danger of oxidation is quite significant. Therefore, antioxidants must be included because oxidation lowers the nutritional and sensory quality of food. In the past, manufacturers added synthetic antioxidants such as butylhydroxyanisole (BHA), butylhydroxytoluene (BHT), and butylhydroxyquinone (BHQT), but for health reasons, some countries and industries banned them and replaced them with vitamin E [13,51]. Despite the benefits of this molecule, it remains a synthetic molecule, raising the question of whether natural extracts may also take its place. To assess the antioxidant capacity of OFI optimum as a natural antioxidant in margarine, an accelerated oxidation test (rancimat) was used. Oxidative stability in this test was expressed as the time required to reach a critical oxidation point (time stability in hours), as shown in Table 4.

According to data presented in Table 4, OFI peel extract as a natural antioxidant increases the margarine’s oxidative stability by 3.2 h, with the induction time equal to 16.02 ± 0.41 h. Better oxidative stability is demonstrated in margarine with OFI peel optimum extract compared to those containing vitamin E. Our extract demonstrated a longer induction time than vitamin E-fortified margarines, 14.33 ± 0.22 and 15.86 ± 0.17, respectively, even at 50 ppm. The effectiveness of an antioxidant in protecting fatty acids, particularly emulsions, against fatty acid oxidation is highly dependent on the nature of the compounds, i.e., free hydroxyl groups and polarity [18]. In the present study, vanillic acid is slightly soluble in water, kaempferol-3-O-rutinoside is slightly soluble in water, and the rest of the compounds identified are sparingly soluble in water, which favors their perfect penetration into the emulsion as well as their good protection of fatty acids from oxidation, especially essential fatty acids, which are highly sensitive to oxidation. Knowing that lipid oxidation involves a very complex set of free-radical reactions that take place between fatty acids and oxygen [52], it is obvious to achieve a good oxidative stability of margarine with OFI peel extract, especially as our extract has already shown a great anti-free radical capacity (as already shown in the DPPH radical test). Referring to the phenolic composition of OFI peel (Table 5), other mechanisms are possible; flavonoids are well known for being effective metal chelators [53]. The oxidative stability results of OFI peel optimum extract are a significant advancement, particularly considering that vitamin E (α-tocopherol) is a potent antioxidant and anti-radical that is well suited to lipid oxidation prevention and protection. According to Choe and Min [40], the effectiveness of antioxidants in reducing free radicals in food depends on the energy of dissociation between oxygen and the phenolic hydrogen bond, which is between 70 and 80 kcal/mol and declines in the δ > γ > β > α-tocopherol order (from a thermodynamic point of view, hydrogen transfer from antioxidants is more favorable when the energy of dissociation of the O-H bond in the antioxidants is low). *Opuntia ficus-indica* peels were also incorporated into margarine by Chougui, Djerroud, Naraoui, Hadjal, Aliane, Zeroual, and Larbat [41]. However, our outcomes are far better (100 ppm of OFI improved the margarine’s oxidative stability by 1.35 h compared to margarine containing vitamin E). This could be because of how the phenolic chemicals were extracted. In this previous study, the traditional extraction approach (maceration) was used, whereas ours involved the use of a microwave followed by optimization of the extraction conditions for the phenolic components. In a similar line, several investigations have verified that microwave-assisted extraction of phenolic compounds is more efficient than maceration when it comes to concentration and extraction time reduction (by 42 to 97.33 percent) [54,55,56].

This investigation joins other studies confirming that natural extracts rich in bioactive compounds can be excellent natural antioxidant additives to margarine, such as Amrane-Abider, Nerin, Cannelas, Zeroual, Hadjal, and Louaileche [13] (OFI seed), Ouahrani, Casal, Bachir-bey and Zaidi [18] (*Moringaoleífera* leaves), Serra et al. [57] (herbs and spices: *Rosmarinus officinalis*, *Curcuma longa*, *Thymus piperella*, and *Thymus vulgaris*), and Martínez-Girón [58] (*Capsicum annuum* peel and pulp, and *Solanum betaceum Cav.* juice).

### 3.3. Anticoccidial Activities

The current situation of avian coccidiosis is alarming because the irrational use of various anticoccidial drugs is becoming ineffective against avian coccidiosis due to the increasing resistance of *Eimeria* species. That is why researchers are scrambling to find the best herbal remedy [27,59]. The present study is part of that context. Figure 4 shows that OFI peel optimum extract has an interesting anticoccidial activity with an oocyst reduction rate of 30.06 ± 0.51 percent after 7 h of treatment. According to our knowledge, there is not enough research on OFI peel anticoccidial efficacy. Hence, to highlight the potential of our investigation, other herbal extracts were examined. Our findings are better than those reported for pulp and leaf olive extracts (25.36% and 5.87%, respectively) [22,60], whereas 60.53 ± 0.38 mg/mL of OFI peel extract destroyed 50 oocysts (Figure 5), which is higher than the reported values of Debbou-Iouknane, Nerín, Amrane-Abider, and Ayad [22], where 192.94 mg/mL of olive leaves destroyed half of the *Eimeria* oocysts isolated from broiler chickens. Moreover, while the present results were better even against *Eimiria* isolated from other animals like rabbits, Murshed et al. [61] revealed that the oocyst inhibition percentages were low (below 50%) using 25–100 mg/mL of *Calotropis procera* leaf.

Despite the interesting results of the present study, they remain inferior to those found in our previous study [27], where extracts of OFI flowers destroyed *Eimeria* oocysts after 7 h by 44.89%. This is probably linked to the phenolic compounds, as OFI flowers are rich in phenolic compounds, particularly flavonoids. Kerboeuf et al.’s [62] mini-review reported that certain flavonoids limit protozoan resistance to other drugs. The results of this study showed a correlation between diclazuril (R^2^ = 0.87) and OFI peel extract (R^2^ = 0.95) concentrations in reducing the oocyst numbers (Figure 5).

Instead of testing natural extracts, Alnassan et al. [63] studied the effect of commercial molecules such as allicin on chicken *Eimeria tenella*, which is known for its antibacterial and antiviral activities. Their findings were higher (56.24%) than those of the present study. In the same light, our finding was lower than the quercetin standard, with the oocyst reduction rate equal to 45.38%. Thymol, carvacrol, and saponins were the other standard molecules that exhibited high anticoccidial activity (the *Eimeria* spp. oocyst reduction rate was superior to 50%) [60,64]. These previous studies confirm the direct effect of bioactive substances on *Eimeria* spp.

Plant extracts with high concentrations of bioactives, particularly polyphenolic compounds, have been shown by Molan et al. [65] to inhibit the enzymes involved in coccidian parasite sporulation. According to Abbas et al. [65], plant extracts have the ability to penetrate both oocyst shell layers and induce a loss of intracellular components, which destroys and softens the parasite’s primary cytoplasmic mass. In a similar vein, Ali et al. [66] and Tanweer et al. [67] reported the crucial role of these compounds in *Eimeria* oocyst destruction by modifying membrane ionization, leading to cell death.

Since one of the objectives of this study was to find an alternative to the synthetic treatment, we proceeded in the same way using diclazuril, which is known for its effectiveness in reducing *Emeria* oocysts (Figure 5). High correlation coefficients of diclazuril and OFI peel extract (R^2^ = 0.87 and R^2^ = 0.95, respectively) were found. Our findings demonstrate that, despite the OFI peel extract’s positive effects, it is still not as effective as diclazuril (oocyst reduction rate: 64.70%, LC_50_: 37.65 mg/mL). However, as shown above (Table 5), OFI peel is rich in phenolic acid and flavonoids; maybe it was necessary to proceed with the purification of these compounds to have better effects since pure molecules or standards often give better results because their functional groups are free.

## 4. Conclusions

The optimization of the extraction procedure for phenolic compounds and antioxidant activities of the *Opuntia ficus-indica* peel extract was satisfactory using a Box–Behnken response surface design. The total phenolic content strongly influences the optimized parameters such as ethanol concentration, extraction time, and microwave power. OFI peel extracts have been found to be a source of natural phenolic compounds, such as isorhamnetin, dihydrokaempferol, and kaempferol-3-O-rutinoside, and exhibit high antioxidant activities. In the present study, it was demonstrated that the addition of OFI peel extract to margarine increased resistance towards oxidation, compared to margarine without antioxidants and margarine containing vitamin E. In addition, it was concluded that OFI peel extract possesses the ability to destroy *Eimeria* spp. from broiler chickens and can serve as an alternative to synthetic anticoccidial drugs. In the future, in vivo investigations are required to assess the efficiency of the OFI peel bioactive compounds in poultry.

## Figures and Tables

**Figure 1 foods-12-04403-f001:**
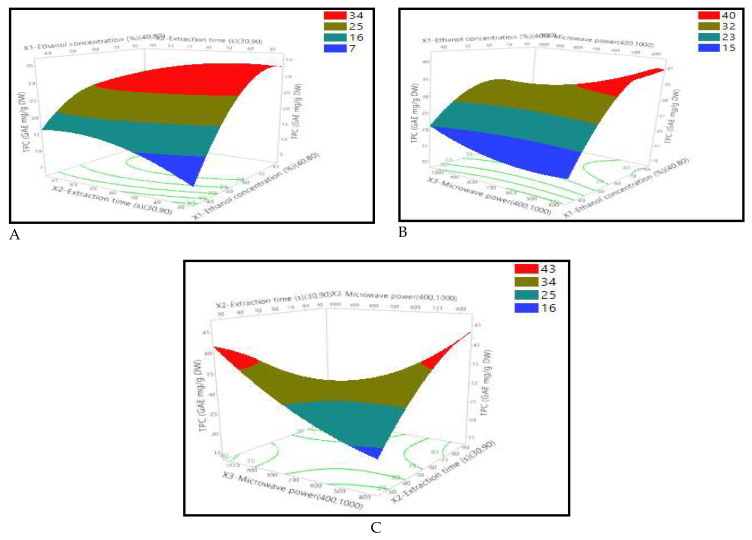
Response surface plots showing the effects of X_1_: ethanol extraction (%), X_2_: extraction time (seconds), and X_3_: microwave power (Watts) on TPC extraction (mg GAE/g DW) from prickly pear peel. (**A**) Represents the interaction plot of ethanol contraction and extraction time factors; (**B**) represents the interaction plot of ethanol contraction and microwave power factors; (**C**) represents the interaction between extraction time and microwave power factors.

**Figure 2 foods-12-04403-f002:**
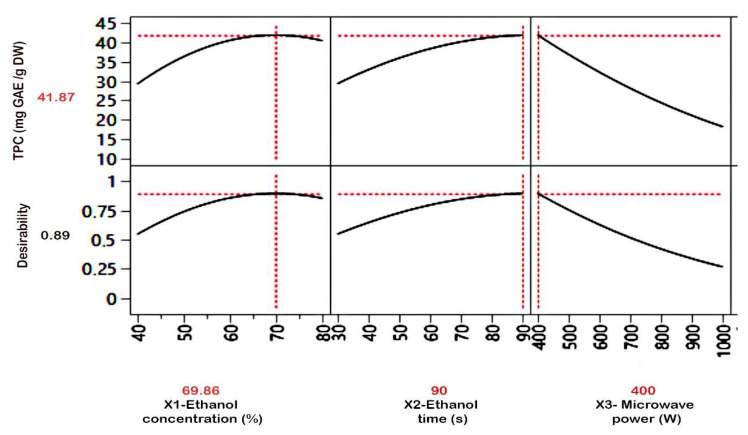
Microwave-assisted extraction parameters and response prediction profile of OFI peel.

**Figure 3 foods-12-04403-f003:**
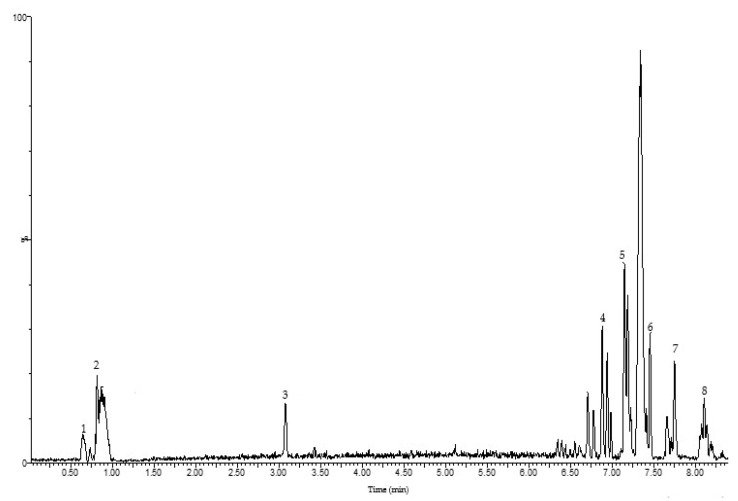
*Opuntia ficus-indica* peels’ optimal phenolic component profile (UPLC-ESI-Q-TOF-MS) (1) Vanillic acid; (2) Coumaric acid; (3) Protocatechuic acid; (4) Isohamnetin; (5) Dihydrokaempferol; (6) Kaempferol-3-O-rutinoside; (7) Isorhamnetin 3-O-rutinoside; (8) Isorhamnetin 3-O-glucoside.

**Figure 4 foods-12-04403-f004:**
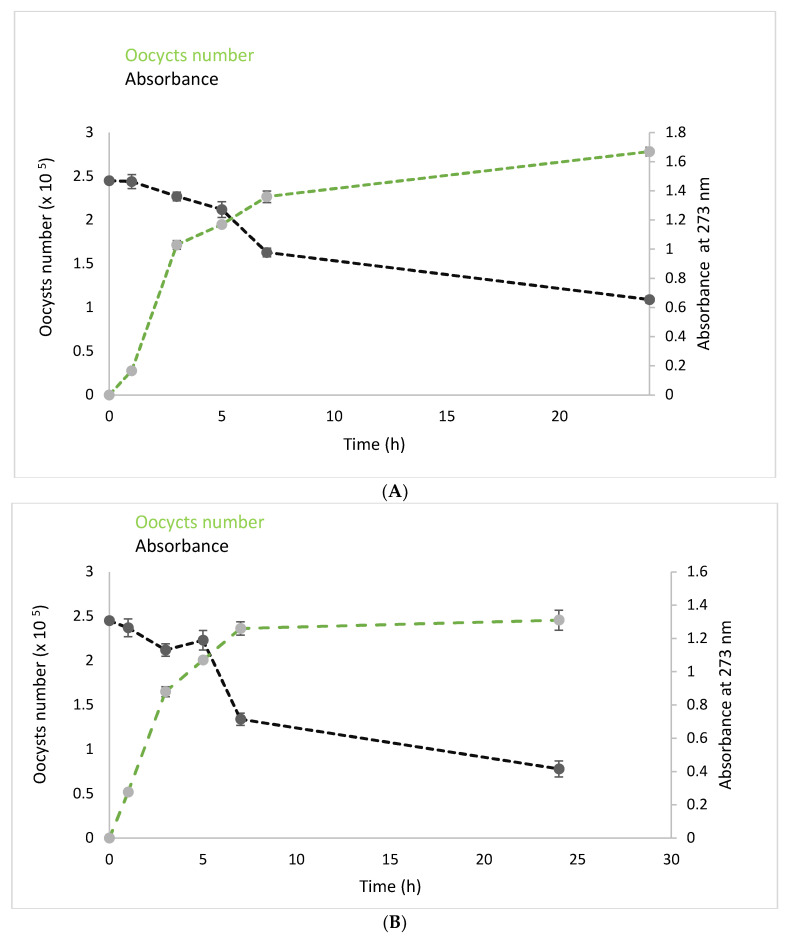
Reduction in the number of *Eimeria* oocysts as a function of time and absorbance at a wavelength of 273 nm after treatment with optimum OFI peel (**A**) and diclazuril (**B**).

**Figure 5 foods-12-04403-f005:**
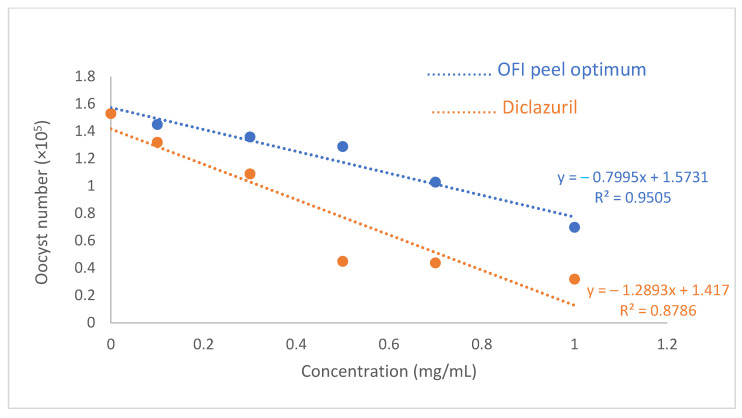
Effect of OFI peel optimum and diclazuril concentrations on oocyst number.

**Table 1 foods-12-04403-t001:** Box–Behnken design matrix experimental plan, experimental data, and predicted values for tree level three extraction factor response surface analysis.

Run	Variable Levels			TPC (GAE mg/g DW)	
	X_1_	X_2_	X_3_	Observed	Predicted
1	40	30	700	12.59	14.47
2	60	60	700	32.32	31.47
3	60	90	1000	19.59	20.12
4	80	90	700	26.97	25.08
5	80	60	400	37.24	38.78
6	80	30	700	34.67	33.70
7	60	60	700	33.82	31.47
8	60	90	400	40.11	40.50
9	60	60	700	29.27	31.47
10	40	60	400	21.66	20.30
11	40	60	1000	25.36	23.86
12	60	30	1000	39.28	38.89
13	80	60	1000	27.94	29.30
14	60	30	400	24.90	24.37
15	40	90	700	19.47	20.44

X_1_: Ethanol concentration (%); X_2_: Extraction time (s); X_3_: Microwave power (W).

**Table 2 foods-12-04403-t002:** Variance analysis (ANOVA) table for the effect of microwave extraction factors (ethanol concentration, time, and power) on TPC extraction (mg GAE/g DW).

Source	DF	Sum of Squares	F Ratio	Prob > F
TPC				
X_1_	1	245.0340	42.5626	0.0013 *
X_2_	1	214.0661	37.1834	0.0017 *
X_3_	1	30.7549	5.3422	0.0688
X_1_X_2_	1	53.1776	9.2370	0.0288 *
X_1_X_3_	1	42.2435	7.3377	0.0423 *
X_2_X_3_	1	304.5200	52.8954	0.0008 *
X_1_X_1_	1	117.9654	20.4907	0.0062 *
X_2_X_2_	1	27.5454	4.7847	0.0804
X_3_X_3_	1	13.2628	2.3038	0.18952
Model	9	886.31	16.71	0.0032 *
Lack of fit	3	18.0061	1.1137	0.5053
Error	5	28.7851		
Total model	14	895.0955		
R^2^ = 0.97				
Adj. R^2^ = 0.90				

Legend: DF, degree of freedom; SS, sum of squares; *, significant influence.

**Table 3 foods-12-04403-t003:** Effect of main factors and interactions.

Variables		*p*-Value
X_2_-Extraction time (s) × X_3_-Microwave power (W)	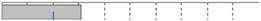	0.00077
X_1_-Ethanol concentration (%)	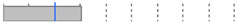	0.00089
X_1_-Ethanol concentration (%) × X_1_-Ethanol concentration (%)	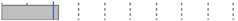	0.00625
X_1_-Ethanol concentration (%) × X_2_-Extraction time (s)	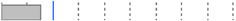	0.02877
X_1_-Ethanol concentration (%) × X_3_-Microwave power (W)	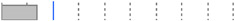	0.04228
X_2_-Extraction time (s) × X_2_-Extraction time (s)	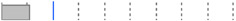	0.08046
X_3_-Microwave power (W)	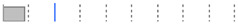	0.14409
X_3_-Microwave power (W) × X_3_-Microwave power (W)	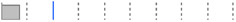	0.18927
X_2_-Extraction time (s)	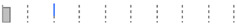	0.47001

Gray bars represent the significance of extraction factors.

**Table 4 foods-12-04403-t004:** Antioxidant activities and margarine oxidative stability of *Opuntia ficus-indica* peel optimum extract. ^a–d^ Different lowercase letters indicate statistically significant differences (*p* < 0.05) according to ANOVA test.

**Antioxidant Activities IC_50_ (mg/mL)**	
DPPH radical	12.99 ± 0.36
Ferric reducing power	6.57 ± 0.05
**Rancimat (hours)**	
Margarine control	12.82 ± 0.51 ^a^
Margarine vitamin E	14.33 ± 0.22 ^b^
Margarine enriched (OFI) 50 ppm	15.86 ± 0.17 ^c^
Margarine enriched (OFI) 100 ppm	16.02 ± 0.41 ^d^

**Table 5 foods-12-04403-t005:** Phenolic compound constituents identified in OFI peel optimum extract by LC-MS/MS.

	Compounds	[M-H]	Retention Time (min)	Concentration (µg/g)	Molecular Formula	Fragment Ions (*m*/*z*)
1	Vanilic acid	167	0.65	60.09 ± 0.13	C_8_H_8_O_4_	167 (100), 153 (5), 137 (3), 123 (1)
2	Coumaric acid	163	0.82	93.12 ± 0.25	C_9_H_8_O_3_	119 (100),163 (22), 91 (2)
3	Protocatechuic acid	153	3.08	81.80 ± 0.19	C_7_H_6_O_4_	153 (39), 109 (100), 111 (29)
4	Isorhamnetin	315	6.88	155.37 ± 0.51	C_16_H_12_O_7_	315 (100), 301 (7), 297 (6), 285 (2)
5	Dihydrokaempferol	287	7.15	206.16 ± 0. 30	C_15_H_12_O_6_	289 (100), 273 (2), 271 (49), 153 (11)
6	Kaempferol-3-O-rutinoside	593	7.46	112.81 ± 0.15	C_27_H_30_O_15_	287 (100), 146 (7)
7	Isorhamnetin 3-O-rutinoside	623	7.78	108.79 ± 0.21	C_28_H_32_O_16_	315 (42), 314 (100)
8	Isorhamnetin 3-O-glucoside	477	8.10	95.80 ± 0.12	C_22_H_22_O_12_	315 (100), 300 (80)

## Data Availability

Data are contained within the article.

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
