# Peer review of "Opuntia Ficus-Indica Peel By-Product as a Natural Antioxidant Food Additive and Natural Anticoccidial Drug"

_foods, 2023, doi:10.3390/foods12244403_

Round 1
Reviewer 1 Report
Comments and Suggestions for Authors
Dear editors,
I have completed the evaluation of the manuscript entitled “Opuntia ficus- indica peel by-product as a natural antioxidant food additive and natural anticoccidial drug” (foods-2726829). This manuscript focuses on the optimizing of phenolic acid extraction from Opuntia ficus-indica peel by-product in Algeria, quantifying and identifying of these phenolic compounds, assessing their antioxidant activity and margarine oxidative stability, as well as evaluating their in vitro anticoccidial efficacy.
In my perspective, this manuscript is interesting and their experimental findings showed a superior performance in comparison to other natural materials. But, there is scope for improvement in terms of presentation and writing style.
Thus, I recommend major revision. Additionally, there are still some questions that need to be answered by the authors. Some specific questions are as follows:
(1) Please clarify the correlation between phenolic compounds and the rate of margarine oxidation.
(2) In the Introduction, the author has already elucidated the antioxidant effects of the Opuntia ficus-india (OFI) peels. What is the significance of assessing antioxidant activity (2.4.2 and 2.4.3) within the experimental design?
(3) In this study, the margarine oxidative stability and in vitro anti-coccidia efficacy were evaluated. Is there any necessary relationship between these two tests?
(4) In line 63, Please provide a detailed description of the advantages of microwave-assisted technology.
(5) Please explain: In Table 3, what do the gray bars represent?
(6) In page 13, line 10, please add some references which states that “This could be because of how the phenolic chemicals were extracted”.
(7) The results should be presented directly in the results section, while the background information can be included in the introduction or omitted (Line 228-232).
(8) The conclusion requires further modification and explanation. It is suggested that the authors not only list the results but also emphasize and summarize their significant findings, while including a concise overview of the limitations of this study and proposing future research directions.
Comments on the Quality of English Language
Minor editing
Author Response
Manuscript ID: foods-2726829
Reviewers' comments:
Reviewer #1:
Dear Editors,
I have completed the evaluation of the manuscript entitled “Opuntia ficus-indica peel by-product as a natural antioxidant food additive and natural anticoccidial drug” (foods-2726829). This manuscript focuses on the optimizing of phenolic acid extraction from Opuntia ficus-indica peel by-product in Algeria, quantifying and identifying of these phenolic compounds, assessing their antioxidant activity and margarine oxidative stability, as well as evaluating their in vitro anticoccidial efficacy.
In my perspective, this manuscript is interesting and their experimental findings showed a superior performance in comparison to other natural materials. But, there is scope for improvement in terms of presentation and writing style.
Thus, I recommend major revision. Additionally, there are still some questions that need to be answered by the authors. Some specific questions are as follows:
Dear reviewer, our sincere thanks for taking the time to review this manuscript, and for your close attention to detail. We highly appreciate your overall positive feedback regarding the quality of the manuscript! Please see below for our responses to your comments:
(1) Please clarify the correlation between phenolic compounds and the rate of margarine oxidation.
Answer: Thank you for that pertinent remark; the correction have been made.
(2) In the Introduction, the author has already elucidated the antioxidant effects of the Opuntiaficus-india (OFI) peels. What is the significance of assessing antioxidant activity (2.4.2 and 2.4.3) within the experimental design?
Answer: Secondary metabolites, which include antioxidant molecules, are typically found in plants. The quantity and quality of these molecules, however, vary depending on the species, climate (a sunny climate is known to promote the synthesis of carotenoids), extraction conditions, and techniques. On the other hand, an increase in the extraction of phenolic compounds does not necessarily lead to an increase in antioxidant activities. It all depends on the nature, quality, and quantity of the bioactive molecules. i.e., in the present research, an evaluation of antioxidant activities is necessary to verify and confirm antioxidant activity and deduce the mechanism of these molecules. Moreover, through these tests, we have confirmed that OFI peel compounds are hydrogen and electron donors.
(3) In this study, the margarine oxidative stability and in vitro anti-coccidia efficacy were evaluated. Is there any necessary relationship between these two tests?
Answer: This study aims to maximize the value of OFI peel by-products in two distinct domains (food additives and alternative veterinary medicine); these two endeavors are linked by the phenolic compounds that play a direct role in each of them.
(4) In line 63, Please provide a detailed description of the advantages of microwave-assisted technology.
Answer: Thank you for that pertinent remark; the correction have been made. Line 66-70
(5) Please explain: In Table 3, what do the gray bars represent?
Answer: Thank you for that pertinent remark; the correction have been made in the revised version.
(6) In page 13, line 10, please add some references which states that “This could be because of how the phenolic chemicals were extracted”.
Answer: The corrections have been made in the revised version.
(7) The results should be presented directly in the results section, while the background information can be included in the introduction or omitted (Line 228-232).
Answer: The corrections have been made (the introduction has been removed).
(8) The conclusion requires further modification and explanation. It is suggested that the authors not only list the results but also emphasize and summarize their significant findings, while including a concise overview of the limitations of this study and proposing future research directions.
Answer: Thank you for this valuable remark; the correction has been made
THANK YOU AGAIN FOR YOUR REVIEW!
Reviewer 2 Report
Comments and Suggestions for Authors
While a good portion of the methodology is similar to that reported by the same authors in "Bioactive Compounds and In Vitro Antioxidant and Anticoccidial Activities of Opuntia ficus-indica Flower Extracts," the information obtained is novel due to the incorporation of antioxidants into margarine and its application to kill Eimeria oocysts. The researchers present a series of experiments in which they have previous experience, so I have only a few comments:
- It is suggested that the use of Opuntia peel be employed to extract antioxidants with ethanol, protecting the environment in the process, and that it can replace treatments with a 30% mortality rate. I believe this reach is overly optimistic. Instead of making a promise, it could be indicated as having potential.
- In paragraph 124, is the experimental triplicate intentional?
- In paragraphs 142 and 143, this section is not clear. Could it be more descriptive? For example, indicate that this is part of the margarine analysis.
- In paragraph 184, was a control run with only the solvent used? What about the death of Eimeria at the final ethanol concentration?
- In paragraph 203, after incubation... under what conditions? How much time?
- In paragraph 310, Figure 1, it lacks an explanation of the colors or the box in the upper right corner. What units are represented?
- Due to space constraints, some paragraphs were omitted to describe the questions. In the results section, regarding the description of phenolic compounds identified in LC-MS/MS, in the last paragraph, it is mentioned that the variability of compounds is due to different conditions in various studies. Is the microwave extraction process better than other reported methods for obtaining sufficient compounds with antioxidant activity?
- In paragraph 378, words need to be separated.
- Was there a taxonomic identification by sequencing of the isolated Eimeria strain?
- Considering cereals as the main food for fattening poultry, is it profitable to use margarine as an additive in the livestock industry?
- If a higher ethanol concentration results in a higher compound yield, is it cost-effective to use large amounts of ethanol to process one ton of Opuntia peel? How much compound should be obtained to completely eliminate Eimeria?
Author Response
Reviewer #2:
While a good portion of the methodology is similar to that reported by the same authors in "Bioactive Compounds and In Vitro Antioxidant and Anticoccidial Activities of Opuntia ficus-indica Flower Extracts," the information obtained is novel due to the incorporation of antioxidants into margarine and its application to kill Eimeria oocysts. The researchers present a series of experiments in which they have previous experience, so I have only a few comments:
Dear reviewer, our sincere thanks for taking the time to review this manuscript, and your close attention to detail. We highly appreciate your overall positive feed-back regarding the quality of the manuscript! Please see below for our responses to your comments:
- It is suggested that the use of Opuntia peel be employed to extract antioxidants with ethanol, protecting the environment in the process, and that it can replace treatments with a 30% mortality rate. I believe this reach is overly optimistic. Instead of making a promise, it could be indicated as having potential.
Answer: According to the anticoccidial tests carried out in vitro with different proportions of ethanol, the expression of oocyst destruction was not observed while supporting the work of our laboratory [1]. Contrary to the results noted by [2], ethanol at 50 and 70% expressed the destruction of the wall of oocysts of the Eimeria species.
- In paragraph 124, is the experimental triplicate intentional?
Answer: We couldn't find the right line for your question, but you can be sure that all the experiments we merge are repeated three times to calculate the average and the ecartype, and to avoid handling errors.
- In paragraphs 142 and 143, this section is not clear. Could it be more descriptive? For example, indicate that this is part of the margarine analysis.
Answer: Thank you for that pertinent remark; the correction have been made (line 178-179)
- In paragraph 184, was a control run with only the solvent used? What about the death of Eimeria at the final ethanol concentration?
Answer: In this study we used ethanol as a control at an optimal concentration of 69.86% (figure 2), and no alteration or destruction of the oocyst wall was observed by microscopic observation.
- In paragraph 203, after incubation... under what conditions? How much time?
Answer: Thank you for that pertinent remark; the correction have been made (line 212-215)
- In paragraph 310, Figure 1, it lacks an explanation of the colors or the box in the upper right corner. What units are represented?
Answer: It's a very relevant suggestion, but unfortunately, the software gives the figure like that (without a unit). We could solve the problem with other software such as Photoshop or Pint, but unfortunately, the resolution of the image always decreases. Normally,with the Z axis, the unit of TPC is clear.
- Due to space constraints, some paragraphs were omitted to describe the questions. In the results section, regarding the description of phenolic compounds identified in LC-MS/MS, in the last paragraph, it is mentioned that the variability of compounds is due to different conditions in various studies. Is the microwave extraction process better than other reported methods for obtaining sufficient compounds with antioxidant activity?
Answer: It's difficult and unfair to compare an optimized method with a non-optimized one. Although microwaves are a fast extraction method, they need to be optimized because high power or long processing times can damage the molecules. As with other methods, the extraction conditions are crucial to the extraction and activity of bioactive compounds.
- In paragraph 378, words need to be separated.
Answer: Thank you for that pertinent remark; the correction have been made
- Was there a taxonomic identification by sequencing of the isolated Eimeria strain?
Answer: As our laboratory is not equipped with sophisticated equipment, basic methods have been borrowed. The identification of Eimeria species in chickens was done on the basis of criteria such as size, shape, presence or absence of apical anatomical structure (micropyle), intestine localization and its sporulation time [3-5].
The proportions of Eimeria species was added in line192-194
- Considering cereals as the main food for fattening poultry, is it profitable to use margarine as an additive in the livestock industry?
Answer: the aim of this study is not to use margarine as an additive in the livestock industry, but to maximize the value of OFI bark by-products in two distinct fields (food additives and alternative veterinary medicine); these two fields are linked by the phenolic compounds that play a direct role in each of them.
- If a higher ethanol concentration results in a higher compound yield, is it cost-effective to use large amounts of ethanol to process one ton of Opuntia peel? How much compound should be obtained to completely eliminate Eimeria?
Answer: In the present investigation, we have optimized the ethanol concentration to extract the maximum amount of phenolic compound. The idea is to extract with water, but unfortunately, these compounds are moderately polar, so an extraction solvent is necessary (in our laboratory, we carried out preliminary studies in which we tested ethanol, water, methanol, and acetone; ethanol proved to be the best). A higher ethanol concentration or another parameter does not necessarily result in a higher yield of compounds, hence the importance of optimization.
According to figure 2, using the optimum conditions, we extracted 41.87 mg GAE/gDW of TPC in just 90 s (1.5 min), while extracting the same quantity using conventional methods required more time and generally less compound due to the oxidation that takes place, especially since the extract contains chlorophyll, which is known to be a prooxidant. In conclusion, on an industrial scale (tons), profitability is not just a question of ethanol concentration but must take into account all parameters such as time and energy consumption.
Unfortunately, our optimuin (41.87 mgGAE/g DW) was unable to eliminate the parasite completely, so other studies are required, such as compound purification and even in vivo studies, which sometimes give very different results from in vitro.
THANK YOU AGAIN FOR THE REVIEW!
Reviewer 3 Report
Comments and Suggestions for Authors
Opuntia ficus- indica peel by-product as a natural antioxidant 2 food additive and natural anticoccidial drug
The research is interesting and the methods and results are well presented. However, I would suggest the authors to include all the relevant chromatograms of LC-MS analysis and also put the calibration curves or calibration equations of the eight identified compounds as supplementary materials.
Lines 28-29 are not clear.
Line 34: Remove word "the" exhibits a high antioxidant potential
Line 136: The DPPH radical
Line 223: Please give a brief explanation of statistical approaches you used.
Section 3.1 Lines 228-237 should be omitted from results and discussion and the authors may wish to add these lines in the introduction.
Line 407: one of the objectives of this was to find, replace is with was
Comments on the Quality of English Language
The English needs minor editing and language corrections.
Author Response
Reviewer #3:
The research is interesting and the methods and results are well presented. However, I would suggest the authors to include all the relevant chromatograms of LC-MS analysis and also put the calibration curves or calibration equations of the eight identified compounds as supplementary materials.
Dear reviewer, our sincere thanks for taking the time to review this manuscript, and your close attention to detail. We highly appreciate your overall positive feed-back regarding the quality of the manuscript! All of the relevant chromatograms of LC-MS analysis of the calibration curves were included in the manuscript as supplementary material. Please see below for our responses to your comments:
Answer: we've added a new Figure 3, containing chromatograms and calibration curves.
- Lines 28-29 are not clear.
Answer: The correction have been made
- Line 34: Remove word "the" exhibits a high antioxidant potential.
Answer: Thank you for that pertinent remark; the correction have been made
- Line 136: The DPPH radical.
Answer: Thank you for that pertinent remark; the corrections have been made
- Line 223: Please give a brief explanation of statistical approaches you used.
Answer: The corrections have been made in the experimental design and statistical analyses sections.
- Section 3.1 Lines 228-237 should be omitted from results and discussion and the authors may wish to add these lines in the introduction.
Answer: The corrections have been made (the introduction has been removed).
- Line 407: one of the objectives of this was to find, and replace is with was.
Answer: The correction has been made.
THANK YOU AGAIN FOR YOUR REVIEW!